# Relationships among Healthcare Providers’ Job Demands, Leisure Involvement, Emotional Exhaustion, and Leave Intention under the COVID-19 Pandemic

**DOI:** 10.3390/healthcare11010056

**Published:** 2022-12-25

**Authors:** Yun-Tao Li, Shi-Jun Chen, Kuo-Jui Lin, Gordon Chih-Ming Ku, Wen-Yang Kao, I-Shen Chen

**Affiliations:** 1Graduate Institute of Sports Training, University of Taipei, Taipei 100234, Taiwan; 2Department of Leisure Industry Management, National Chin-Yi University of Technology, Taichung 411030, Taiwan; 3Department of Physical Education, University of Taipei, Taipei 100234, Taiwan; 4Department of Health Industry Technology Management, Chung Shan Medical University, Taichung 402306, Taiwan; 5Office of Physical Education, National Chin-Yi University of Technology, Taichung 411030, Taiwan

**Keywords:** job demands, leisure involvement, emotional exhaustion, leave intention

## Abstract

The COVID-19 pandemic has caused many medical issues. It has tested the impact of healthcare providers’ job demands, emotional exhaustion, and other pressures related to the impact on organizational leave intention. Accordingly, the purpose of this study was to verify the relationship between healthcare providers’ job demands, leisure involvement, emotional exhaustion, and leave intention under the COVID-19 pandemic. The questionnaire survey was used to address the issue of the present study. Convenience sampling was utilized to recruit 440 healthcare providers with a validity rate of 95%. Collected data were analyzed by structural equation modelling. Results indicated that healthcare providers’ job demands do not significantly influence leisure involvement. Job demands significantly influence emotional exhaustion. Job demands significantly influence leave intention. Emotional exhaustion significantly influences leave intention. Emotional exhaustion has a significant mediating effect between job demands and leave intention. Finally, relevant practical suggestions are provided based on the study results.

## 1. Introduction

Since 2019, the COVID-19 pandemic has been spreading globally for the past three years. With the emergence of mutant strains, mass vaccinations have been carried out worldwide. However, the pandemic continues to persist, and the number of infected cases and deaths continues to increase. By 2022, the number of confirmed infections globally exceeded 300 million, and deaths exceeded 5 million. Although the world has gradually moved toward the stage of coexisting with the virus, the medical systems of various countries are still facing resulting pressure. Specifically, with the rapid increase in the demand for diagnosis and treatment, the burden on the medical system is becoming heavy, and healthcare providers are facing unprecedented pressure. As [1] proposed, burnout is a key medical care problem during COVID-19 due to the heavy workload of healthcare providers. Regarding physiological and psychological stressors, healthcare providers may also experience ambivalence because they are worried about infecting their relatives and friends due to caring for patients. This will further increase healthcare providers’ work stress, which can lead to their decision to leave/quit.

The study by [2] found that in dimensions of physiology, society, and organization, to achieve tasks, workers’ continuous physiological and psychological efforts are usually accompanied by physiological and psychological costs, which are what the researchers define as “job demands.” Healthcare providers have been working on the front lines for a long time, and their workload has increased significantly since the outbreak of the pandemic. It has become commonplace to work overtime continually due to emergent measures. Research regarding the mental state of healthcare workers presented by [3] has further indicated that nurses are the group with the heaviest load and the most physically and mentally exhausted. The pressure caused by the pandemic often leads to an increase in nurses’ turnover intention in countries around the world. This may worsen the imbalance of the supply and demand of healthcare resources. The psychological state of healthcare workers in high-pressure working environments and relevant issues triggered by such psychological states have been explored before the COVID-19 pandemic. For example, [4] conducted a two-year study with 69 physicians responsible for palliative care and found that physicians are at risk of emotional exhaustion due to the pain and death they face every day; common symptoms include alexithymia, high stress, low perceived social support, and burnout. The authors of [5] also studied the effects on healthcare workers in palliative care who often face ethical decisions surrounding end-of-life and death, leading to physical, psychological, and emotional distress and work-related stress. It is evident that healthcare providers are exposed to considerable stress on a regular basis, which has been exacerbated by COVID-19. During the COVID-19 pandemic, various epidemic control measures and the lack of medical resources have affected the working environment and work content of healthcare providers. Therefore, healthcare providers’ job demands under long-term pressure need to be further explored.

As mentioned above, frontline healthcare providers are working under heavy pressure due to COVID-19. In addition to reducing social contact, they also lack time for leisure due to longer working hours. Participation in leisure activities is a behavior that is helpful and meaningful to people through shared connections. By participating in leisure activities, people can benefit both physically and mentally. A study by [6] suggests that sports can reduce the severity of personal fatigue, work fatigue, and benefit the mental health of healthcare providers. Therefore, this study’s second motivation was to explore healthcare providers’ current leisure involvement.

In the frontline medical scene, healthcare providers usually need to switch between various emotions. Under the current COVID-19 pandemic, healthcare providers are faced with many emotional situations such as aging and death in addition to being required to maintain a good attitude during interactions with patients. The process of accompanying the patients and family members through the COVID-19 pandemic will inevitably impact healthcare providers’ emotions. The authors of [7] highlighted that while healthcare workers provide professional service, they must often hide or suppress their emotions. Healthcare workers often need to coordinate their minds and emotions and bury their feelings to meet their professional roles of work, which is typical of emotional labor. As such, healthcare providers’ emotional resource consumption and ability to regulate the emotions of others are also worth studying. The study by [8] suggested that emotional exhaustion is the most apparent symptom of job burnout and work stress. When workers present emotional exhaustion, their psychological resources have been exhausted, and they can no longer contribute more. Therefore, the third motivation of this study is to explore the current situation of the emotional exhaustion of healthcare providers.

As the pandemic situation changes, healthcare providers are under continual pressure to work on the frontlines and are more likely to leave medical organizations or relevant units. The authors of [9] indicated that due to the current COVID-19 pandemic, workers of medical organizations are under greater pressure than ever before. Further, different pressures are creating various results at the organizational level, including voluntary turnover rates. Under the COVID-19 pandemic, the high turnover rate of healthcare providers has become one of the primary focuses of the medical system. The study by [10] identified that leave intention means that workers want to leave their current workplace due to associated physical or mental health problems. However, whether the reasons are personal factors, work variants, or external factors need to be further explored. Based on the above findings, this study proposed five research hypotheses: (1) job demands have a negative impact on leisure involvement; (2) job demands positively impact emotional exhaustion; (3) job demands positively impact leave intention; (4) emotional exhaustion positively impacts leave intention; (5) emotional exhaustion has a mediating effect on job demands and leave intention. This study recruited healthcare providers as the research subjects and carried out quantitative research. The collected data were analyzed with AMOS 23.0 statistical software. 

Based on the relevant literature, it is evident that under the COVID-19 pandemic, research regarding healthcare providers primarily focused on turnover intention. For example, the study by [11] on the correlation between the pressure of Saudi healthcare providers and turnover intention indicated that pressure is related to turnover intention. The study by [12] recruited healthcare providers in Pakistan providing treatment to COVID-19 patients as subjects and explored the correlation between the perceived risks of COVID-19 with turnover intention. The results found that the healthcare providers’ perceived risks of the COVID-19 pandemic enhanced their fear of COVID-19 and their turnover intention. However, few studies were conducted regarding the leave intention of healthcare providers. With the ongoing pandemic, medical capacity depends on a sound management system and personnel working in the healthcare system. Therefore, the purpose of this study is to examine whether healthcare providers want to give up their current work due to physical or psychological problems. Leave intention is taken as the topic for discussion, which compares with the research results of turnover intention and provides medical system managers with a better understanding of the work needs and leisure involvement of healthcare providers. Therefore, this study investigates leave intention as the primary research topic. In addition to comparing the results of employee turnover intention, this study also provided the medical system managers with a further understanding of the healthcare providers’ job demands and leisure activities.

## 2. Literature Review

It has been known for a long time that the high satisfaction of workers in the workplace is often due to their strong demand and active involvement in their jobs. The study by [13] found that job demands are the source of mental stress. For example, the need for a large amount of work to be performed quickly, without enough time, and the need to deal with conflicts are all factors causing individuals workplace stress. The study by [14] also had similar findings. They defined job demands as projects that need completing and can be taxing both physically and mentally. Job demands are related to the physical and mental workload of employees. Based on this definition, the work nature and environmental factors may cause healthcare providers increased pressure due to emotions, work-family conflicts, and workload, further increasing their job demands.

The author of [15] regarded job demands as one of the primary sources of stress in research related to job demands. In his proposed job demands model, he showed that sources of workplace stress are from the work environment and are also affected by the combination of job demands and job control. The study by [2] proposed the job demand resource model. They highlighted that job demands are the physical and mental costs paid by workers and that work resources can be considered factors that improve the motivation of workers [2]. Work fatigue and work motivation are both developed in the workplace through two different psychological processes. Work fatigue occurs from excessive job demands, while rich work resources can increase work motivation. Therefore, it is evident that job demands are mostly regarded as physical and mental costs. The study by [16] also found that excessive job demands (excessive workload) will exhaust workers’ spirit and physical strength and cause mental exhaustion and health problems.

The concept of involvement is one of the key topics in behavioral research, especially after adding psychological variables. It is more helpful to clarify people’s leisure behavior. The study by [17] noted that relevant research has changed the concept of involvement from individual behaviors to leisure and recreation, suggesting that leisure involvement can be a variety of leisure participation decisions inspired by specific activities or particular environments. The authors of [18] showed that leisure involvement referred to the psychological state such as an excited or focused state that individuals feel when participating in their favorite leisure activities. The correlation can inspire individuals to have an unconscious motivation or interest through the stimulation of specific situations. The authors of [19] described this from the perspective of individuals’ various inputs into leisure activities. They pointed out that regarding leisure involvement, the time, money, and relevant equipment invested in a certain leisure activity is the degree of interest of participants in their leisure activities. Based on the above definition, the leisure involvement of healthcare providers can be described as their participation degree in different leisure activities. The degree is also reflected in various influencing factors, such as equipment, time, and frequency.

Regarding the measurement of leisure involvement, [20] put forward the Involvement Profile as early, which mainly includes four dimensions of importance, pleasure, symbolism, and risk, to explore the factors that leisure activities bring individuals from these four dimensions. The authors of [21] defined leisure involvement in five dimensions, including core value, importance, pleasure, interest, and self-expression. Integrating ego involvement with relevant leisure research has attracted extensive attention and interest in academia. The authors of [22] proposed various factors of activity involvement from the social psychology dimension: (1) attraction: importance and pleasure relevant to the activity; (2) centrality: value and activity (i.e., work) relevant to other fields of life; (3) self-expression: expression of personal identity after participating in the activities. Thus, the concept of leisure involvement provides an understanding of the motivation of healthcare providers to participate in certain leisure activities and a further understanding of healthcare providers’ energy, frequency, and equipment invested in a certain activity. With this in mind, leisure involvement can help to understand the combination of the degree of leisure involvement and work attitudes of healthcare providers from emotion, cognition, and behaviors.

The study by [23] found that emotional exhaustion is a state of psychological reaction to the individuals’ perceived exhaustion of their emotional resources. In other words, for jobs with a relatively high demand for emotional labor, a person’s emotions cater too much to job demands, resulting in emotional exhaustion. The study by [24] discussed burnout in three dimensions, including emotional exhaustion, depersonalization, and reduced personal accomplishment. Emotional exhaustion is characterized by a lack of energy and physical exhaustion. Due to the gradual accumulation of emotional demands at work, an individual’s long-term reaction to work-related stress accumulation will eventually lead to burnout. Depersonalization refers to the negative reactions of individuals at work when facing customers, such as passive attitude and indifference. A lack of accomplishment refers to the situation that individuals hold a negative evaluation of work efficiency and their work skills, which can lead to poor work performance [25].

The study by [26] suggested that when individuals interacting with others at work are required to present expected emotions beyond their load, emotional exhaustion will occur. This will further lead to severe negative behaviors such as depersonalization and reduced personal accomplishment. From the relevant definitions above, emotional exhaustion means a state of physical exhaustion and emotional exhaustion. The study by [27] mentioned that under emotional exhaustion, workers will feel demoralized and unable to devote themselves to working due to the feeling of being exploited and eventually lose a sense of accomplishment for work. Emotional exhaustion influences many areas, as shown in the research conducted by [28], where emotional exhaustion had a complete mediating effect on work-family conflicts and job satisfaction. In this study, an increase in work-family conflicts was found to directly lead to higher emotional exhaustion and further lower the job satisfaction of the project workers [28]. Relevant research also describes the factors influencing emotional exhaustion. For example, [29] discovered that workload and supervisor support are significant antecedents of emotional exhaustion, while emotional exhaustion significantly influenced the turnover intention of the hotel workers in their study, which further lowered service performance.

Intention to quit has always been an important factor in understanding employee turnover [30]. Relevant studies attempted to explore employees’ intention to quit through turnover intention. The author of [31] highlighted that employee turnover intention is a complicated process; It encompasses a negative psychological reaction to specific occupations or organizational conditions, which can further develop into “quit behavior” such as leaving or possible action to give up one’s current job and seek better employment opportunities [31]. The concept of leave intention is similar to turnover intention, and most of the studies on organizational leave intention are based on the intentions concept of [32]. The authors of [10] pointed out that leave intention occurs due to physical or mental problems, where the workers want to give up their current work and choose to quit. Lian mentioned that leave intention is the most effective predictor of turnover behavior. The study by [33] proposed that most researchers focused on attitudes relevant to work and employment alternatives to explain the workers’ leave intention. In other words, few relevant studies were conducted regarding the impact of job demands and emotional exhaustion on leave intention. The study by [34] suggested that in comparison to the actual turnover behavior, more attention should be paid to the direct or indirect influencing factors of the workers’ leave intention.

The relevant literature indicates that research regarding work stress and emotional exhaustion of healthcare providers has continually been an important research topic. Taking the healthcare professionals working in the fields of anesthesiology and emergency as the research subjects, [35] explored whether work-related risk factors increase the risk of burnout. They found that burnout is likely to occur for healthcare professionals working in the fields of anesthesiology and emergency. The researchers also discussed that risk factors can increase the risk of burnout, a finding that is useful for developing remedial health policies. 

Based on the relevant literature, it is evident that studies have focused on psychological or psychological topics regarding healthcare providers under the COVID-19 pandemic. For example, [36] analyzed the impact of the COVID-19 crisis on healthcare providers’ mental health. The results showed that most problems can be put into five categories: stress, depression, the anxiety of being infected by the virus, pain and insomnia, post-traumatic stress syndrome, and suicide. Further, studies also showed how many factors lead to anxiety, depression, and stress, such as the fear of being infected and spreading it to family and friends, stress switch, and lack of rest. The study by [37] explored the psychological influence of the COVID-19 pandemic on healthcare providers; they found that the mental stress of workers in the respiratory department was the highest, and the mental stress is higher in the areas with the highest incidence rate of COVID-19.

The study by [38] used semi-structured interviews to study 24 healthcare providers in Japan and found the main problems faced by healthcare providers under the COVID-19 pandemic were: (1) increased stress and loneliness, (2) reduced coping strategies, (3) communication and acknowledgment as a mental health resource, and (4) understanding self-care. In addition, according to the psychological resilience study of healthcare providers conducted by [39], researchers found that to improve the healthcare providers’ psychological resilience during the COVID-19 pandemic, the sleep quality, positive emotions, and life satisfaction of healthcare providers require improvements. The authors of [39] also found that healthcare providers in their later years had higher psychological resilience. It can be seen that relevant studies on healthcare providers’ physical and mental problems under the COVID-19 pandemic were conducted. However, few studies were conducted on how to make adjustments and what strategies should be taken for the physical and mental issues they have encountered. Relevant studies have addressed the physiological and psychological problems faced by healthcare providers under the COVID-19 pandemic; however, there is little research reporting on how they can adjust and what strategies they should adopt. Therefore, this study explored the issues faced by healthcare providers and also discussed how to invest in leisure to improve the current situation.

## 3. Methodology

### 3.1. Research Structure

This study explored the relationship between healthcare providers’ job demands, leisure involvement, emotional exhaustion, and leave intention under the COVID-19 pandemic. The research structure was based on relevant literature, as shown in Figure 1.

### 3.2. Research Hypotheses

(1) Job demands have a negative impact on leisure involvement.

(2) Job demands have a positive impact on emotional exhaustion.

(3) Job demands have a positive impact on leave intention.

(4) Emotional exhaustion has a positive impact on leave intention.

(5) Emotional exhaustion has a mediating effect on job demands and leave intention.

### 3.3. Research Subjects

Recruiting healthcare providers as the research subjects, this study recruited subjects using the convenience sampling method. From 1 May to 1 July 2022, questionnaires were distributed among nurses and emergency medical treatment workers at National Taiwan University Hospital, China Medical University Hospital, Taitung Hospital, Sin-Lau Hospital, Changhua Christian Hospital, and Pingtung Hospital. The questionnaires were distributed on the scene or by Google questionnaire websites using social community software. A total of 500 questionnaires were distributed, and 460 were collected with a recovery rate of 92%. After excluding the invalid questionnaires, a total of 440 valid questionnaires were obtained, and the validity rate was 95%.

### 3.4. Research Tools

Based on the relevant literature and adopting the framework of the questionnaire by [10], this study’s questionnaire was distributed to healthcare providers. The questionnaire was divided into five parts and 42 items, including nine basic personal data items, 10 job demands items, 11 leisure involvement items, five emotional exhaustion items, and seven leave intention items. This study was based on a five-point Likert scale, and each item was given a score of 1 to 5 points from “strongly disagree” to “strongly agree,” respectively.

### 3.5. Data Processing and Analysis

This study used SPSS Version 23.0 to record the valid questionnaires’ responses and then used AMOS Version 23.0 to analyze the correlation between variables.

## 4. Research Results

### 4.1. Sample Characteristics

According to Table 1, this study involved a total of 440 subjects. Most were females aged 31 to 40, unmarried, and had completed education at the college level. They were primarily residents in central Taiwan, with work experience of four to six years as non-managerial employees. Typically, their level of working overtime on weekdays and holidays was considered an ordinary level.

### 4.2. Measurement Mode Analysis

In terms of the validity and reliability of the questionnaire, this study conducted a test through confirmatory factor analysis and referred to indexes of modification indices (M.I.) for item correction [40]. Based on M.I., this study deleted the D2 of the job demands scale; C2, C6, and C10 of the leisure involvement scale, and A1 and A5 of the leave intention scale.

This study conducted a convergent validity test regarding the dimensions of job demands, leisure involvement, emotional exhaustion, and leave intention. The results show that the factor load of all dimensions was between 0.76 and 0.94, the composite reliability was between 0.91~0.95, and the average variance extracted was between 0.67~0.81, which indicates that this study had convergent validity, as shown in Table 2, Table 3, Table 4 and Table 5.

### 4.3. Discriminant Validity

The discriminant validity of this study was calculated by Bootstrap sampling with a correlation coefficient 95% confidence interval between the dimensions (Table 6 and Table 7). If there is no value of 1 in the correlation coefficient 95% confidence interval, it means that there is good discriminant validity between the dimensions [41].

### 4.4. Test of Mediating Effect

Bootstrap sampling was used to estimate the standard error, and the significance level of mediating effect was calculated. As shown in Table 8, the total effect standard error of job demands to leave intention was 0.507, the lower limit value of 95% CI based on the percentile method was 0.625, and the upper limit value was 0.381. The lower limit value of the bias-corrected percentile method was 0.378, and the upper limit value was 0.623. The confidence interval of the percentile method is 95%, and the bias-corrected percentile method does not contain 0, so the total effect is supported, which indicates that there may be indirect effects. Further tests on indirect effects were conducted.

Table 8 shows that the indirect effect standard error of job demands to leave intention was 0.435, the lower limit value of 95% CI based on the percentile method was 0.301, and the upper limit value was 0.578. The lower limit value based on the bias-corrected percentile method was 0.306, and the upper limit value was 0.589. The confidence interval of the percentile method was 95%, and the bias-corrected percentile method does not contain 0, so the indirect effect is supported, which indicates that the mediating effect exists [42].

### 4.5. Structural Mode Analysis

This study referred to the research of [43,44,45] and used the χ^2^ test, the ratio of χ^2^ and degree of freedom, GFI, AGFI, RMSEA, CFI, and PCFI, totaling seven indicators to test the overall mode fitness (Figure 2). From Table 9, it can be seen that after correction, χ^2^ is 827.26, the ratio of χ^2^ to the degree of freedom is 2.62, GFI is 0.90, AGFI is 0.90, RMSEA is 0.06, CFI is 0.96, and PCFI is 0.86, indicating that the analysis results of this mode are acceptable.

### 4.6. Discussion

The empirical analysis shows that the two hypotheses of work demand on emotional exhaustion and emotional exhaustion on leave intention are both significant. Under the COVID-19 pandemic, the psychological state of health providers and related emotional issues need to be further explored, which also illustrates the importance of the results of this study. The research results are discussed in this paper as follows: 

As shown in Table 10, H1 is not supported. The analysis results show that job demands positively impact leisure involvement, which is consistent with the results found by [45]. The possible reasons may be that healthcare providers bear a heavy workload and are under high pressure, so they want to balance the workload by engaging in leisure activities.

H2 is supported. Job demands have a positive impact on emotional exhaustion. The result is consistent with the study by [46]. The possible reason is that in addition to providing medical service and care, healthcare providers simultaneously need to worry about the effects caused by the pandemic, increasing their emotional labor.

H3 is not supported. The results show that job demands have no positive impact on leave intention, which is the same as that of [10]. The possible reason is that due to personal career or considerations such as economic factors, the leave intention may not necessarily be increased although job demands are high.

H4 is supported. Emotional exhaustion has a positive impact on leave intention. This result is similar to the conclusions of [47]. The reason may be that healthcare providers face a high degree of emotional labor, which will have different degrees of impact on physiology, psychology, and social life over a long time, and will further lead to the desire to leave the medical care profession.

H5 is supported. The analysis results show that emotional exhaustion has a mediating effect on job demands and leave intention. This result is consistent with that of [10], where, in addition to the direct impact on healthcare providers’ leave intention, job demands can also impact healthcare providers’ leave intention through a mediating effect on emotional exhaustion. Therefore, the higher the healthcare providers’ job demands, the higher the healthcare providers’ emotional exhaustion, which will also increase the healthcare providers’ leave intention.

## 5. Conclusions

This study explored the relationship between healthcare providers’ job demands, leisure involvement, emotional exhaustion, and leave intention under the COVID-19 pandemic. Taking the healthcare providers as research subjects, this study conducted a survey questionnaire through convenience sampling. Data analysis found that under the COVID-19 pandemic, healthcare providers’ job demands had no significant negative effect on leisure involvement. However, healthcare providers’ job demands significantly positively impacted emotional exhaustion and leave intention. Healthcare providers’ emotional exhaustion had a significant impact on leave intention. Finally, according to the test of the mediating effect, healthcare providers’ mediating effect of emotional exhaustion on job demands and leave intention have reached a significant level.

## 6. Recommendations

### 6.1. For Healthcare Providers

The results of this study show that healthcare providers’ job demands had a positive impact on leisure involvement. Therefore, it is suggested that healthcare providers could invest enough in leisure activities to switch emotions in environments with a high degree of emotional labor. For example, healthcare providers working for long periods indoors could participate in outdoor sports in their spare time as compensation. Through changing environments, their five senses would be immersed in the outdoors, which will benefit them mentally, emotionally, and physically. Many activities such as jogging, hiking, and swimming can be carried out alone. Through leisure participation and self-perceiving processes, they can learn more about their emotions. At the same time, the process of experiencing outdoor activities through leisure involvement can also help relieve stress.

On the other hand, in the process of leisure involvement, community support from participating in specific activities will also help mediate job demands. People are more likely to receive social support by participating in outdoor group activities. Getting to know people with different lives, cultures, and beliefs through shared activities, and through telling and listening during interpersonal interactions, will be helpful to release the pressure from workplace stress.

The results of this study also show that healthcare providers’ emotional exhaustion had a positive impact on leave intention. It is suggested that healthcare providers could understand their career planning and adjust their attitude in a timely manner. When facing work stress and high levels of emotional exhaustion/gloom, they can recall their original intentions in wanting to work in healthcare and find the value in that work again. At the same time, they can conduct self-assessments, such as an overall subjective assessment of their current health situation. If the individuals are in an uncomfortable state in the workplace and still believe they are healthy psychologically, then it can be interpreted as a state of self-perceived good health. In addition, getting moderate rest and exercise can help to properly cope with the pressure. In this way, it is possible to reduce the leave intention, engage in health-promoting behaviors, and keep a positive attitude in the workplace.

The results of this study also show that healthcare providers’ emotional exhaustion had a mediating effect on job demands and leave intention. When job demands were stronger, emotional exhaustion became larger and led to a higher leave intention. Therefore, it is suggested that healthcare providers consider their workload to be happy at work in a relaxed mood, reduce emotional exhaustion, effectively manage stress, and reduce their leave intentions.

### 6.2. For Medical Institution Managers

The results of this study show that job demands had a positive impact on emotional exhaustion. Therefore, it is suggested that under the current pandemic, when the overall environment seems difficult to change, facing the anxiety and job demand of the healthcare providers, it is up to the medical institution managers to provide more psychological support to healthcare providers and further reduce the healthcare providers’ emotional exhaustion. Regarding specific measures, when human resources are permitted, activities such as regular ward meetings or staff dinners could be held, and professionals or senior medical colleagues could also be invited to share work experiences with healthcare workers. EAP could also be taken to invite professionals to provide long-term and systematic service plans. Personal consultation could be provided regularly to healthcare providers, assisting them to solve their problems at this stage, and helping them to reduce stress. Senior medical colleagues could also be invited to share work experiences with the healthcare workers to further understand the expectation of the healthcare providers in the workplace, providing proper support and channels to slow down the generation of emotional exhaustion.

The results of this study show that job demands have no positive impact on leave intention. Nevertheless, medical institution managers can still focus on establishing appropriate stress management channels for healthcare providers, such as holding relevant outdoor sports lectures to cultivate correct leisure attitudes and concepts, holding relevant courses on leisure treatment and consultation, or even providing colleagues with experience in participating in outdoor sports. These are all helpful to reduce the leisure constraints, improve their willingness to participate in leisure, adjust the work stress on the scene, and relieve the stress load to achieve a balanced and relaxing effect.

## 7. Limitations of the Study and Research Contribution

This study recruited healthcare providers in Taiwan as the research subjects. Although the subjects were located in hospitals in northern, central, and southern Taiwan, the conclusions of this study have limitations when inferring to other countries. Taiwan’s pandemic prevention measures are relatively strict, and pandemic prevention regulations have not yet been fully relaxed. Compared to many other countries that are shifting towards coexisting with the virus, Taiwan’s stricter pandemic prevention measures may put more pressure on the medical system; hence, the healthcare providers are still in a high-pressure work environment. However, the results of this paper indicated that working in a highly stressful environment, in addition to the proper arrangement of medical manpower and the input of relevant resources through the management policies of medical institutions, the intervention of leisure activities presented another adjustment strategy for healthcare providers. Specifically, the benefits brought by leisure activities to healthcare providers are often positive. The perspective of leisure involvement introduced by this study can inject new avenues of future research. This study contributes to the larger body of research by suggesting the importance of improving the work environment and mood of healthcare providers while reducing their work stress.

Moreover, through a comprehensive review of relevant studies on healthcare providers under the COVID-19 pandemic, this research found that most studies are quantitative research focusing on quality measurement produced by a few variables, such as the quantitative research on work stress, job satisfaction, emotional labor, and employee turnover. However, few studies were conducted on self-perceived health topics. As self-perceived health is a description of evaluating one’s own health status, although some healthcare providers are in a highly stressful working environment, their self-perceived health is better than that of other healthcare providers. Hence, they have higher motivation to participate in activities to maintain their healthy behavior. These deeper psychological feelings are often unable to be measured by a scale. Future suggestions on healthcare providers’ related research topics can be analyzed for perceived health.

## Figures and Tables

**Figure 1 healthcare-11-00056-f001:**
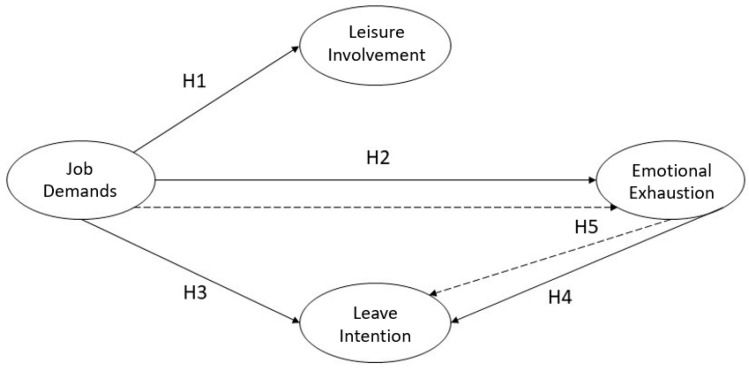
The research structure diagram.

**Figure 2 healthcare-11-00056-f002:**
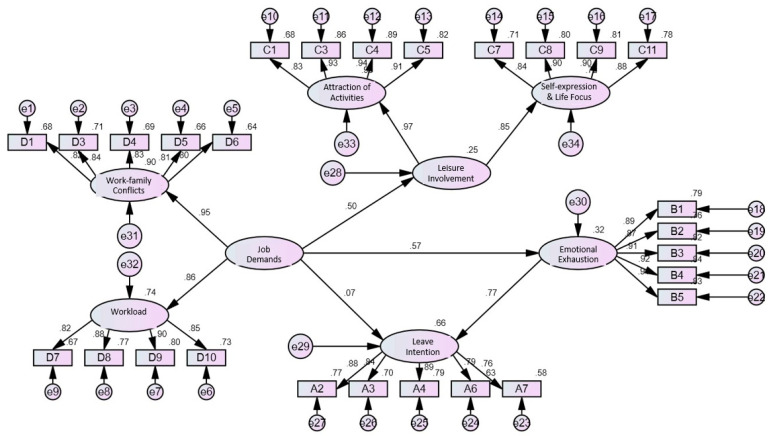
Relation diagram of healthcare providers’ job demands, leisure involvement, emotional exhaustion, and leave intention under the COVID-19 pandemic.

**Table 1 healthcare-11-00056-t001:** Characteristics of Subjects.

Background Variables	Classification Standard	Number of Samples	Percentage	Accumulative Percentage
Gender	Male	160	36.4	36.4
	Female	280	63.6	100.0
Age	30 years old (including) and below	135	30.7	30.7
	31–40 years old	181	41.1	71.8
	41–50 years old	97	22.0	93.9
	51 years old (including) and above	27	6.1	100.0
Marital status	Married	169	38.4	38.4
	Single	261	59.3	97.7
	Other	10	2.3	100.0
Educational level	Vocational and senior high school (including below)	42	9.5	9.5
	College (including a five-year junior college program)	334	75.9	85.5
	Graduate institute (including above)	64	14.5	100.0
Residence	Northern Taiwan	65	14.8	14.8
	Central Taiwan	249	56.6	71.4
	Southern Taiwan	98	22.3	93.6
	Eastern Taiwan	22	5.0	98.6
	Offshore islands	6	1.4	100.0
Job position	Managerial employee	93	21.1	21.1
	Non-managerial employee	347	78.9	100.0
Job tenure	Under 2 years	34	7.7	7.7
	2–4 (under) years	50	11.4	19.1
	4–6 (under) years	110	25.0	44.1
	6–8 (under) years	91	20.7	64.8
	8–10 (under) years	69	15.7	80.5
	Over 10 years	86	19.5	100.0
Do you often work overtime on weekdays?	Often	140	31.8	31.8
	Normal	225	51.1	83.0
	Not often	75	17.0	100.0
Do you often work overtime on holidays (weekends or national holidays)?	Often	126	28.6	28.6
	Normal	213	48.4	77.0
	Not often	101	23.0	100.0

**Table 2 healthcare-11-00056-t002:** Summary of Convergent Validity and Dimension Reliability-Job Demands.

Dimension	Index	Standardized Factor Loading	C.R.	AVE
Work-family conflict	D1	0.82	0.91	0.67
	D3	0.85		
	D4	0.83		
	D5	0.81		
	D6	0.80		
Workload	D7	0.82	0.92	0.74
	D8	0.88		
	D9	0.90		

**Table 3 healthcare-11-00056-t003:** Summary of Convergent Validity and Dimension Reliability-Leisure Involvement.

Dimension	Index	Standardized Factor Loading	C.R.	AVE
Activity attraction	C1	0.83	0.94	0.81
	C3	0.93		
	C4	0.94		
	C5	0.91		
Self-expression Life Focus	C7	0.84	0.93	0.77
	C8	0.90		
	C9	0.90		
	C11	0.88		

**Table 4 healthcare-11-00056-t004:** Summary of Convergent Validity and Dimension Reliability-Emotional Exhaustion.

Dimension	Index	Standardized Factor Loading	C.R.	AVE
Emotional exhaustion	B1	0.88	0.95	0.80
	B2	0.87		
	B3	0.91		
	B4	0.92		
	B5	0.91		

**Table 5 healthcare-11-00056-t005:** Summary of Convergent Validity and Dimension Reliability-Leave Intention.

Dimension	Index	Standardized Factor Loading	C.R.	AVE
Leave intention	A2	0.87	0.91	0.69
	A3	0.85		
	A4	0.89		
	A6	0.79		
	A7	0.76		

**Table 6 healthcare-11-00056-t006:** Job Demands Bootstrap with Correlation Coefficient 95% Confidence Interval.

			Bias-Corrected			Percentile Method	
			Estimate	Lower bound	Upper bound	Lower bound	Upper bound
Work-family conflict	<-->	Workload	0.83	0.76	0.87	0.77	0.87

**Table 7 healthcare-11-00056-t007:** Leisure Involvement Bootstrap with Correlation Coefficient 95% Confidence Interval.

			Bias-Corrected			Percentile Method	
			Estimate	Lower bound	Upper bound	Lower bound	Upper bound
activity attraction	<-->	Self-expression/Life Focus	0.82	0.74	0.87	0.74	0.87

**Table 8 healthcare-11-00056-t008:** Summary of Mediating Effect.

	Estimate	95% Confidence Interval		
Indirect effect		BC/PC *p* value	BC	PC
job demands ->emotional exhaustion ->leave intention	0.435	0.001/0.001	0.306-0.589	0.301-0.578
Direct effect				
job demands ->emotional exhaustion	0.567	0.001/0.001	0.419-0.696	0.424-0.704
job demands ->leave intention	0.072	0.289/0.305	−0.053–0.240	−0.055–0.237
emotional exhaustion ->leave intention	0.767	0.001/0.001	0.582–0.888	0.586-0.890
Total effect				
job demands ->leave intention	0.507	0.001/0.001	0.378–0.623	0.381–0.625

BC: Bias-corrected percentile method; PC: Percentile method

**Table 9 healthcare-11-00056-t009:** Overall Fitness Analysis of the Research Mode.

Fitness Indicator	Acceptable Criteria	After Mode Revision	Mode Fitness Judgment
χ^2^ (Chi-square)	The lower, the better.	827.26	
Ratio of *χ^2^* to the degree of freedom	<3	2.62	Suited
GFI	>0.80	0.90	Suited
AGFI	>0.80	0.90	Suited
RMSEA	<0.08	0.06	Suited
CFI	>0.80	0.96	Suited
PCFI	>0.50	0.86	Suited

**Table 10 healthcare-11-00056-t010:** Summary of hypotheses and validation results.

Hypotheses	Path Value	Validation Result
H1: Job demands have a negative impact on leisure involvement.	0.50	Not supported
H2: Job demands have a positive impact on emotional exhaustion.	0.57 *	Supported
H3: Job demands have a positive impact on leave intention.	0.07	Not supported
H4: Emotional exhaustion has a positive impact on leave intention.	0.77 *	Supported
H5: Emotional exhaustion has a mediating effect on job demands and leave intention.		Supported

* *p* < 0.05.

## Data Availability

Data is not accessible. According to the informed consent of the study, all participants’ personal information and transcripts have to be confidential. Therefore, data cannot be made publicly available.

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
