# Peer review of "Relationships among Healthcare Providers’ Job Demands, Leisure Involvement, Emotional Exhaustion, and Leave Intention under the COVID-19 Pandemic"

_healthcare, 2022, doi:10.3390/healthcare11010056_

Round 1
Reviewer 1 Report
Thank you for submitting the manuscript. I read your paper with great interest. The topic is certainly relevant, but in my opinion revisions are needed to make it fully readable.
In section 2 there are several typos regarding punctuation and capital letters: please correct.
The passage about emotional exhaustion is confusing and ungrammatically written. Emotional exhaustion is one of the three dimensions that make up Burnout, not an extension of it. Please clarify more rigorously.
Information about the psychological situation of health care workers before the pandemic should be included in sections 1 and 2. In fact, reading your paper it seems as if the problem was only the pandemic. Instead, there is ample literature that shows that the pandemic has exacerbated, unmasked and highlighted already existing problems.Therefore, please enter the following references that will also help you in the drafting of the additional paragraphs: doi: 10.13075/ijomeh.1896.01302. doi: 10.11124/JBISRIR-2016-2633. DOI: 10.3390/healthcare10081370
Please remove the words "recommendations" and, if you feel it is necessary, rephrase "suggestions". Yours is a study, not a guideline.
I would like to enter the protocol number of the Ethics Committee, and the Ethics Committee that authorized the study. Also I would like to know if the study has been registered.
The whole paragraph about the contributions of the authors, the funds etc. is missing. Please also include information about funds etc.
I hope my comments are useful to you.
Kind Regards
Reviewer 2 Report
Abstract: should be written again in academic forms. Main aim, where study conducted, model used, period, and main findings.
I propose in the introduction should specify the methodology of research and research hypotheses. The diagnosis itself should indicate the novelty of the results and to publish the considerations in scientific journals. It should define the purpose of the work and its significance, including specific hypotheses being tested. The current state of the research field should be reviewed carefully and key publications cited.
The introduction should include the structure of the paper and its purpose.
Does not understand this record (see Literature review, e.g., lines: 107, 116). Why is there a period after a note, then a note again:
For example, the need for a large amount 105 of work to be done quickly, without enough time, and the need to deal with conflicts are 106 all factors causing individuals workplace stress [11]. [12] also had similar findings.
... and job control [13]. [2] proposed the job demand resource model.
Please review the entire Literature Review section.
In Literature Review: The research gap must be created by a systematic literature review that provides 'holes' in the state of knowledge on the topic. At the end of the justification you should write something like: According to what we were able to find, there are no studies referring and reporting on ... With this you have therefore proven that the issue is relevant, and you have also proven that your study does indeed fill a research gap.
Conclusion is optional, consider dropping the conclusion and create a chapter Discussion.
It is a pity that the chapter was abandoned: Discussion. This would probably have shown what our results mean in general and why our analyses are important.
What did we establish new in our research?
What did others know, and what do we know?
What are the similarities and differences in the results?
What conclusions can be drawn from this?
What research plans do we have?
Did our results confirm the hypothesis?
In conclusion, I propose
-evaluate the critical research, show its limitations and weaknesses,
- highlight the new knowledge and the lessons learned from it,
- describe the importance of the research and how it affects the wider field, show how the information obtained can be further used
Conclusions must be clearly and unambiguously linked to the results of the survey. Their theoretical and practical implications should be indicated (This section is not mandatory but can be added to the manuscript if the discussion is unusually long or complex.)
Round 2
Reviewer 1 Report
Thank you for submitting the manuscript. I have read with great interest and attention the revisions you have made to your manuscript and I am convinced that it is now worthy of publication. Well done.
Kind Regards
Reviewer 2 Report
The authors adapted the manuscript to the reviewer's suggestions. I recommend that the article be accepted for publication in its present form.